# AP205 VLPs Based on Dimerized Capsid Proteins Accommodate RBM Domain of SARS-CoV-2 and Serve as an Attractive Vaccine Candidate

**DOI:** 10.3390/vaccines9040403

**Published:** 2021-04-19

**Authors:** Xuelan Liu, Xinyue Chang, Dominik Rothen, Mariliza Derveni, Pascal Krenger, Salony Roongta, Edward Wright, Monique Vogel, Kaspars Tars, Mona O. Mohsen, Martin F. Bachmann

**Affiliations:** 1International Immunology Center, Anhui Agricultural University, Hefei 230036, China; 2Department of Immunology, RIA, Bern University Hospital, 3010 Bern, Switzerland; xinyue.chang@dbmr.unibe.ch (X.C.); dominik.rothen@students.unibe.ch (D.R.); pascal.krenger@students.unibe.ch (P.K.); salony.roongta@dbmr.unibe.ch (S.R.); Monique.vogel@dbmr.unibe.ch (M.V.); martin.bachmann@me.com (M.F.B.); 3Department of BioMedical Research, University Bern, 3012 Bern, Switzerland; 4School of Life Sciences, University of Sussex, Brighton, BN1 9QG UK; m.derveni@hvivo.com (M.D.); ew323@sussex.ac.uk (E.W.); 5Latvian Biomedical Research & Study Centre, Ratsupites iela1, LV 1067 Riga, Latvia; kaspars@biomed.lu.lv; 6Saiba AG, 8808 Pfäffikon, Switzerland

**Keywords:** virus-like particles, AP205-VLPs, RB motif, humoral immune response

## Abstract

COVID-19 is a novel disease caused by SARS-CoV-2 which has conquered the world rapidly resulting in a pandemic that massively impacts our health, social activities, and economy. It is likely that vaccination is the only way to form “herd immunity” and restore the world to normal. Here we developed a vaccine candidate for COVID-19 based on the virus-like particle AP205 displaying the spike receptor binding motif (RBM), which is the major target of neutralizing antibodies in convalescent patients. To this end, we genetically fused the RBM domain of SARS-CoV-2 to the C terminus of AP205 of dimerized capsid proteins. The fused VLPs were expressed in *E. coli*, which resulted in insoluble aggregates. These aggregates were denatured in 8 M urea followed by refolding, which reconstituted VLP formation as confirmed by electron microscopy analysis. Importantly, immunized mice were able to generate high levels of IgG antibodies recognizing eukaryotically expressed receptor binding domain (RBD) as well as spike protein of SARS-CoV-2. Furthermore, induced antibodies were able to neutralize SARS-CoV-2/ABS/NL20. Additionally, this vaccine candidate has the potential to be produced at large scale for immunization programs.

## 1. Introduction

The current global pandemic of COVID-19 caused by the new coronavirus SARS-CoV-2 has a large impact on the economic and social life worldwide. To date, about 120 million people have been infected, with approximately 2.5 million deaths (source:WHO). Similar to the disease caused by the other two viruses SARS (SARS-CoV) and MERS (MERS-CoV), COVID-19 leads to systemic infection and may cause organ failure, particularly of the lung [1]. In contrast to SARS-CoV and MERS-CoV, which could be contained more efficiently, SARS-CoV-2 spreads more rapidly because of infected asymptomatic individuals (mostly pre-symptomatic) who can transmit the virus [2]. To limit the damage of COVID-19, primary efforts focus on confinement, with physical distancing and multiple further non-pharmaceutical measures to prevent infection [3,4]. At present, many investigations aim at defining optimal strategies to limit viral transmission while simultaneously permitting business and social life activities [5]. One of these strategies is to rapidly increase the human immunity using vaccines against COVID-19. Different vaccines based on RNA and adenoviral vectors have been developed and several of them have already been approved for emergency use in many countries. Other traditional vaccine platforms such as those based on recombinant proteins or subunit vaccines are still in clinical or preclinical development [6]. Despite the achievement of marketed vaccines, there is still a strong need for more doses as re-vaccinating people into 2022 may be required as well as having a platform for future emerging coronaviruses. Therefore, investing in the development of safe, effective, and affordable coronavirus vaccines continues to be of high priority.

The neutralizing antibodies against SARS-CoV-2 are mostly specific for the receptor binding domain (RBD) of the spike protein. Within RBD, the receptor binding motif (RBM) is considered the most important site which directly interacts with ACE2. As compared to SARS-CoV, the affinity to ACE2 of the original spike of SARS-CoV-2 is about 4-fold higher, offering an explanation for the increased infectivity of the latter. Consequently, generating a vaccine inducing antibodies against spike or RBD is the preferred strategy of the majority of COVID-19 vaccine candidates [7]. It has recently been shown that in contrast to most parts of the spike protein, RBD shows minimal glycosylation as it only contains two glycosylation sites, and the RBM is not glycosylated at all. Furthermore, the two glycosylation sites of RBD are distant from the RBD-ACE2 interaction interphase (RBM) and do not interfere with the spike-ACE2 receptor interaction [8]. This confirms that the generation of vaccines inducing antibodies against RBD and especially against RBM is a valuable strategy to fight the pandemic [7]. 

Virus-like particles (VLPs) represent one of the relatively traditional vaccine platforms in the sense that they have already proven their clinical usefulness. VLPs consist of viral structural proteins that upon recombinant expression, self-assemble into particles, mostly icosahedrons and rarely helical [9]. We have previously described the favorable characteristics of VLPs and the reason they constitute a good vaccine template and have given rise to highly successful vaccines [10,11,12]. VLPs derived from Leviviridae have gained considerable attention in the field of vaccinology as a versatile platform [13]. A great interest has been directed against the capsid architecture of phage-VLPs such as MS2, PP7, Qβ and recently AP205. Such knowledge is crucial to pinpoint the surface reactive amino acids which are essential to chemically couple or insert foreign epitopes within the coat protein (CP). AP205 phage infects Acinetobacter bacteria. AP205-VLPs forms stable particles and have a robust coat protein that can accommodate large insertions into the C or N termini [13]. The crystallographic asymmetric unit of an AP205-VLP contain a single CP dimer which consists of chains A and B representing two monomers. The dimer structure feature is special in as much as the C terminus of one monomer is in close proximity with the N-terminus of the other monomer. In comparison to other phages such as MS2, the CP of AP205 is missing a beta strand in its N-terminus and has an extra beta strand in the C terminus. Accordingly, AP205 has N and C-termini in the same location and well-exposed on the surface which explains the ability of AP205 to tolerate and accommodate long insertions at either termini without compromising the capsid integrity [14]. 

In this study, we have shown that the bacteriophage AP205-VLPs can be used as an effective platform for the development of a COVID-19 vaccine against SARS-CoV-2. This system also facilitates large-scale production of the vaccine. The developed platform can also be optimized to tackle the emerging variants of SARS-CoV-2, an area of research we are currently exploring. 

## 2. Materials and Methods

### 2.1. Mice

All in vivo experiments were performed using (8–12 weeks old) wild-type female BALB/cOlaHsd mice purchased from Harlan. All animal procedures were conducted in accordance with the Swiss Animal Act (455.109.1–5 September 2008) of University of Bern. All animals were treated for experimentation according to protocols approved by the Swiss Federal Veterinary Office.

### 2.2. AP205-RBM Vaccine Cloning, Expression, and Production

To design and develop AP205-RBM vaccine, we first genetically fused two codon-optimized synthesis AP205 monomers and subcloned them into the modified pETDuet-1 vector to construct the pETDuet-1-AP205 dimer plasmid. RBM gene encoding residues 437–508 of SARS-CoV-2 Spike protein (GenBank accession number QIA98606.1) was amplified by polymerase chain reaction (PCR )with Q5^®^ High-Fidelity Master Mix (New England Biolabs, Ipswich, MA, USA) using codon-optimized synthesis pUCIDT-SARS-CoV-2-RBD plasmid gifted from Prof Andris Zeltins (Latvian Biomedical Research and Study Center) as template and C-terminal fused with VLP AP205-dim between the *Bmt* I and *Hind* III sites in the pETDuet-1-AP205 dimer plasmid from our laboratory and a C-terminal 6xHis-tag. For amplification following primers were used for PCR: F: 5′-tctgatactactgctagcggatccaacagcaacaacc-3′ and R: 5′-attatgcggccgcaagctttagtgatggtgatggtgatgactagtatacggctgatag-3′. The corresponding PCR fragment was analyzed in 1.2% agarose gel and then purified with *GeneJet Gel Extraction* kit (Thermo Scientific, Carsbad, CA, USA). The PCR product of 3′terminal end and plasmid pETDuet-1-AP205 were digested with enzymes *Bmt* I and *Hind* III (Thermo Fischer Scientific, Waltham, MA, USA) and ligated, resulting in plasmid pETDuet-1-AP205-RBM. *E. coli* XL1-Blue cells were used as a host for cloning and plasmid amplification. After sequencing, the plasmid clones without sequence errors were then subcloned into T7 Express Competent *E. coli* C2566 (High Efficiency) (New England Biolabs, Ipswich, MA, USA).

Expression and purification of AP205-RBM: *E. coli* cultures were grown in lysogeny broth (LB) medium containing Ampicillin (100 μg/mL) on a rotary shaker (200 rev/min; Infors, Bottmingen, Switzerland) at 37 °C to an OD600 of 0.4–0.8. Then, the expression was induced with 0.1 mM Isopropyl-β-D-thiogalactopyranoside (IPTG). Incubation was continued on the rotary shaker at 16 °C for 16 h. The resulting biomass was collected by low-speed centrifugation and was frozen at −70 °C. After thawing on ice, the cells were suspended in the buffer containing 20 mM Tris-HCl pH 8.0 supplemented with 100 mM NaCl with additional 2 mM EDTA, 1 mM PMSF, 5% glycerol, and 0.1% Triton X-100 were disrupted by ultrasonic treatment. Insoluble and soluble proteins were separated by centrifugation (14,000 rpm, 20 min at 4 °C). All steps involved in the expression of VLP were monitored by SDS-PAGE using 12% gels.

### 2.3. Protein Refolding and Purifying

Cell pellets was resuspended in lysis buffer (10 mL buffer for 1 g cells), sonicated and centrifuged for 20 min, 10,000× *g* at 4 °C. Discard supernatant, the pellets were resuspended for individual wash steps: sonicate, centrifuge for 20 min, 10,000× *g* at 4 °C, and repeated for 4×. After last time centrifuge, debris was resuspended in inclusion body (IB) solubilization buffer (8 M urea, 20 mM Tris-HCl and 100 mM NaCl) and incubate for 16 h, 4 °C on rotating wheel (slow rotation). Following centrifuging for 20 min, 10,000× *g* at 4 °C, IB supernatant was dialyzed against refolding buffer (RB) I buffer with 4 M urea, 20 mM Tris-HCl, 0.5 M Arginine, 5 mM reduced Glutathione and 0.5 mM L-oxidized Glutathione for 24 h at 4 °C, subsequently RB II buffer (2 M urea, 20 mM Tris-HCl, 0.5 M Arginine, 5 mM reduced Glutathione and 0.5 mM L-oxidized Glutathione) for 24 h at 4 °C. Finally, dialyzed in RB III buffer (20 mM Tris-HCl, 0.5 M Arginine, 5 mM reduced Glutathione and 0.5 mM L-oxidized Glutathione) for 36 h at 4 °C. After centrifuge for 20 min, 10,000× *g* at 4 °C, VLP refolding fusion protein in supernatant was analyzed on 12% SDS-PAGE gel and was purified by HisTrap™ High Performance (GE Healthcare, Freiburg, Germany).

### 2.4. Electron Microscopy

2 μL of purified AP205-RBM protein (1 mg/mL) suspension for negative staining were adsorbed on glow discharged and carbon coated copper grids (Plano, Wetzlar, Germany) for 1 min at room temperature (RT). After 3× washing by pure water, grids were stained with 2% uranyl acetate solution (Electron Microscopy Science, Hatfield, MA, USA) for 30 s. The excess fluid was removed by gently pushing them sideways to filter paper. Samples were then examined with a transmission electron microscope (Tecnai Spirit, FEI, Hillsboro, OR, USA) at 80 kV and equipped with a digital camera (Veleta, Olympus, Münster, Germany).

### 2.5. Mass Spectrometry

AP205-RBM (1 mg/mL) were diluted with a 3-hydroxypicolinic acid matrix solution and were spotted onto a modulated tool path (MTP) Anchor Chip 400/384TF. Matrix-assisted laser desorption/ionization (MALDI)-TOF MS analysis was carried out on a MALDI-MS (Thermo Scientific, San Jose, CA, USA). 

### 2.6. Vaccination Regimen

Wild-type Balb/c female mice (8 weeks, Harlan) were vaccinated subcutaneously (s.c.) with 100 µg AP205-RBM in 100 µL PBS or with AP205 VLPs control in 100 µL PBS on day 0. Mice were boosted with a similar dose on day 28. Serum samples were collected on days 14, 21, 35, and 49, respectively. 

### 2.7. The Enzyme-Linked Immunosorbent Assay (ELISA) 

For determination of total IgG antibodies titers against AP205-RBM in sera of immunized mice, ELISA plates were coated with 0.1 µg/mL and 1.0 μg/mL spike RBD or full spike proteins (Sinobiological, Beijing, China) overnight, respectively. Plates were washed with PBS-0.01% Tween and blocked using 100 µL PBS-Casein 0.15% for 2 h in RT. Sera from immunized mice were diluted 1/20 initially and a 1/3 dilution chain was performed. Plates were incubated for 1 h at RT. After washing with PBS-0.01%Tween (PBST), goat anti-mouse IgG conjugated to Horseradish Peroxidase (HRP) (Jackson ImmunoResearch, West Grove, PA, USA) was added 1/2000 and incubated for 1 h at RT. Plates were developed and OD 450 reading was performed. Antibody titers were determined as the concentrations of antibody at 50% maximum optical density (OD50). IgG subclasses were measured from day 35 sera using the same ELISA protocol with the following secondary Abs: goat anti-mouse IgG1-HRP and goat anti-mouse IgG2a-HRP (1:1000) (Thermo Fischer Scientific, Waltham, MA, USA), goat anti-mouse IgG2b-HRP (SouthernBiotech, Birmingham, AL, USA) 1:4000, rat anti-mouse IgG3-HRP (Becton, Dickinson, Franklin Lakes, NJ, USA) 1:2000 incubated at RT for 1 h.

### 2.8. Antibody Avidity Measurement

To test IgG antibody avidity against SARS-CoV-2 full spike protein and RBD protein, serial three-fold dilution serum samples from immunized mice, 1/20 initially performed, were added to ELISA plates coated over night with 1.0 µg/mL RBD and Spike proteins, respectively. After incubation at RT for 1 h, the plates were washed once in PBS-0.01% Tween, and then washed 3× with 7 M urea in PBS-0.05%Tween or with PBS-0.05% Tween for 5 min every time. After washing with PBS-0.05%Tween, goat anti-mouse IgG conjugated to Horseradish Peroxidase (HRP) (Jackson ImmunoResearch, West Grove, PA, USA) was added 1/2000 and incubated for 1 h at RT. Plates were developed and read at OD 450 nm.

### 2.9. Neutralization Assay

Vesiculoviral pseudotyped virus production has been described elsewhere [14] with the only change being the use of pCAGGS_SARS-CoV-2_Spike (a kind gift from Dr Emma Bentley NIBSC). After establishing the TCID_50_ of the stock on HEK293T/17 cells transiently expressing ACE2 and TMPRSS2 (pCAGGS ACE2 and TMPRSS2 both kind gifts from Dr Nigel Temperton University of Kent), neutralization assays were undertaken using 100 × TCID_50_ per well of 96-well plate. The virus was incubated for 1 h at 37 °C (5% CO_2_) along with the heat-inactivated serum, which was diluted over a range of 1:20–1:500, after which 2 × 10^4^ HEK293T/17 cells transiently expressing ACE2 and TMPRSS2 were added to each well and the plate left to incubate (37 °C, 5% CO_2_) for a further 48 h. The media was then discarded and the level of report gene activity assessed using a 50:50 mix of non-supplemented media: BrightGlo and a read in a GloMax Discover (Promega, London, UK).

CPE-based assay: the capacity of the induced antibodies in neutralizing wild-type SARS-CoV-2 (SARS-CoV-2/ABS/NL20) was also performed. Serum samples were heat-inactivated for 30 min at 56 °C. Two-fold serial dilutions were prepared starting at 1:20 up to 1:160. 100 TCID50 of the virus was added to each well and incubated for 37 °C for 1 h. The mixture has been added on a monolayer of Vero cells and incubated again for 37 °C for 4 days. Four days later the cells were inspected for cytopathic effect (CPE). The titer was expressed as the highest dilution that fully inhibits formation of CPE. 

### 2.10. Statistical Analysis

Data were analyzed and presented as mean ± SEM using GraphPad PRISM 8. Comparison between the groups was performed using Student’s *t*-test (non-parametric). The *p* value was considered statistically significant if <0.05 (** *p* < 0.01; * *p* < 0.05).

## 3. Results

### 3.1. The Bacteriophage AP205-VLPs Can Be Used Efficiently for Generating a Fusion Vaccine Against SARS-CoV-2

In our previous studies, we have provided a proof-of-concept that AP205-VLPs constitute an efficient vaccine platform that is suitable for fusing epitopes of different lengths such as peptides from angiotensin II, CXCR4 receptor, influenza A M2 protein or gonadotropin releasing hormone (GnRH). These epitopes could efficiently be fused at the N or C terminus of AP205 and allowed the assembly of VLPs without compromising their integrity [15]. Furthermore, a stabilized version of AP205 has recently been designed by fusing 2 capsid proteins with a small linker, resulting in VLPs with 90 N and C termini. In the current study, the RBM domain (aa437–508) of the spike of SARS-CoV-2 was added to the C terminus of AP205 dimer as illustrated in Figure 1A. The construct was expressed in *E*. *coli* for the possibility to scale up and allowed spontaneous packaging of ssRNA which serve as toll-like receptor (TLR) 7/8 agonist. Expression resulted in large amounts of insoluble aggregates which could, however, easily be denatured and refolded using urea (Figure 1B). The production of AP205-RBM fusion protein was confirmed by SDS-PAGE showing a band of ~37.4 kDa (Figure 1C) equivalent to AP205 dimers displaying 1 RBM domain. To confirm the integrity of the produced vaccine we performed electron microscopy (EM) as well as Mass Spectrometry (MS) analysis. The EM and MS confirmed the formation of VLPs ~25–30 nm in size as shown in Figure 1D and E. The developed vaccine incorporates ~90 RBM corresponding to the number of dimers in AP205 (Figure 1F).

### 3.2. Vaccination with AP205-RBM Vaccine Induces High Titers of RBD and Spike-Specific IgG Abs 

To test the immunogenicity of the developed vaccine candidate, we performed in vivo studies. BALB/c mice were primed s.c. with 100 μg of AP205-RBM on day 0 and received a boost of 100 μg on day 28 (Figure 2A). Mice were bled weekly to measure the induced antibodies against RBD and Spike proteins of SARS-CoV-2. AP205 VLPs were used as a control. The results showed an increase in RBD (Figure 2B,C) and Spike (Figure 2D,E) specific antibody titers 14 days after priming which continued to increase steadily. Following the booster injection on day 28, the titer increased by ~2-folds. No RBD-specific antibodies were detected in the mice vaccinated with AP205 as a control. 

### 3.3. AP205-RBM Vaccine Promotes IgA Production and IgG Responses Dominated by IgG2a

In a next step, we tested the ability of the vaccine to induce immunoglobulin class-switching to IgA. To this end, ELISA plates were coated with RBD and mouse sera from day 49 was used to measure induction of specific IgA. The obtained results demonstrate a strong ability of the vaccine to induce RBD-specific IgA antibodies as shown in Figure 3A. Measuring the OD50 of IgA response of AP205-RBM confirmed induction of strong IgA responses (Figure 3B). 

Mouse IgG antibodies are known to be heterogenous and basically consist of four subclasses, for BALB/c mice: IgG1, IgG2a, IgG2b, and IgG3. The subclasses vary in their immunological, biochemical as well as physiological properties. It has been shown in previous studies that IgG2a and IgG2b play an essential role against pathogens by enhancing opsonization, complement fixation and immune effector functions [16,17]. Thus, it is of significant interest to understand and compare the induced subclasses in response to vaccination with AP205-RBM.

Our results showed that vaccination with AP205-RBM enhanced the class-switching to IgG2a and IgG1. Induction of IgG2b and IgG3 could also be detected but to a much lesser extent (Figure 3C–E).

### 3.4. The Induced IgG Antibodies against RBD and Spike Protein by AP205-RBM Vaccine Are of High Avidity

The avidity of antibodies can be defined as the binding strength through more than one point of interaction. Avidity can be quantified as the ratio of *K*_d_ for the intrinsic affinity over the one for functional affinity [18]. Nevertheless, avidity term can be exchanged in a looser sense with functional affinity [19]. Modified immunoassays can be used to determine the avidity of the induced antibodies after vaccination. This is carried out by using 7 M Urea washes to detach low avidity antibodies. To this end, ELISA plates were coated with RBD or Spike protein and sera from the terminal bleeding on day 49 were used for further analysis. After incubation with serum dilutions, half the ELISA plates were washed using PBST + 0.05% Tween 20 while the other half was washed using 7 M Urea in PBS + 0.05% Tween 20 as explained in the method section. The results indicate that the induced antibodies against RBD or Spike proteins are of high avidity (Figure 4A,C). When measuring the OD50 of RBD or Spike-specific antibodies, the small differences between the groups washed with PBST or Urea were confirmed (Figure 4B,D), indicating the high avidity of the induced antibodies by the vaccine candidate. To corroborate these findings, we tested the sera of vaccinated mice in ELISA plates coated with very low concentration of RBD (0.10 µg/mL), where we also obtained strong signal, indicating good affinity/avidity of the induced antibodies (Figure 4E,F).

### 3.5. The AP205-RBM Vaccine Candidate Induces Antibodies Neutralizing SARS-CoV2 

The best correlates of protection are virus neutralization for most viral infections and most likely also for SARS-CoV-2. Accordingly, we performed two types of neutralization assays, one using pseudotyped Vesicular Stomatitis Virus with SARS-CoV-2 spike, as well as “real” SARS-CoV-2. For the pseudotyped virus neutralization assay, pseudo-types with the S glycoprotein of SARS-CoV-2 and luciferase for quantification were used to evaluated serum for neutralizing activity against the spike protein. Neutralization activity of the immunized mouse sera were assessed and directly demonstrated the anti-viral neutralizing activity of the induced antibodies (Figure 5A). For the neutralization assay with SARS-CoV-2, we used the CPE neutralization method using 100 TCID50 of SARS-CoV-2/ABS/NL20. Also, in this assay, the induced antibodies we were neutralizing could completely inhibit viral replication (Figure 5B). 

## 4. Discussion

Our first attempt to generate a COVID-19 vaccine was based on producing a recombinant RBD in human eukaryotic cells and chemically coupling the domain to CuMV_TT_-VLPs, resulting in a vaccine candidate that binds to ACE2 and induces neutralizing antibodies in mice [20]. Chemical coupling techniques have proven efficacious for the production of numerous vaccine candidates [21,22,23,24,25]. However, to facilitate GMP-compatible production at a large-scale, genetic fusion techniques may be considered a better option. Purification of recombinantly produced VLPs may be performed at a large scale using continuous flow, ultracentrifugation which is currently used for producing 85% of the world’s influenza vaccine supplies [26]. Hence, the vaccine candidate detailed in this study based on stabilized AP205 VLPs, may offer advantages in terms of large-scale production compared to the previously described one. 

It has recently been shown that in contrast to most parts of the spike protein, RBD shows minimal glycosylation as it only contains two glycosylation sites, while the RBM is not glycosylated at all. Furthermore, the two glycosylation sites of RBD are distant from the RBD-ACE2 interaction interphase (RBM) and do not interfere with the Spike-ACE2 receptor interaction [8]. The fact that RBM, which contains the actual ACE2 binding site, is not glycosylated indicates that the virus cannot exploit glycan shielding to escape immune response without attenuating its infectivity [27]. Importantly, bacterial expression systems may therefore be suitable for productions of RBM-based vaccines, as glycosylation is not of importance. Hence, generation of vaccines inducing antibodies against RBD and especially against RBM in bacteria is a valuable strategy to fight the pandemic [7]. Accordingly, we have chosen RBM as our target and fused it with AP205 stabilized by fusing two capsid proteins into a covalent dimer. This resulted in the formation of immunogenic VLPs displaying RBM on the surface of AP205. AP205-RBM vaccine induced a high antibody titer against both RBD and Spike protein of SARS-CoV-2 even after one dose of the vaccine, a response that could be further enhanced by a booster immunization 4 weeks later.

The vaccine candidate not only induced high levels of antibodies, but they were also of high avidity. Indeed, washes with 7 M Urea removed only a relatively small fraction of antibodies, indicating that the overall response was of high avidity. Some recent studies have shown that antibody responses induced by SARS-CoV-2 infection are relatively weak and of rather short duration, particularly in individuals who had no or only moderate symptoms [28,29]. We have proposed recently that coronaviruses employ a structural strategy to avoid the induction of strong antibody response [30]. Other viruses can induce optimal neutralizing long-lived antibody responses owing to their repetitive surface of typically 60 or more rigid epitopes, spaced by 5–10 nm which are recognized as a pathogen-associated structural pattern (PASP) (AP205 VLPs have this characteristic). In contrast, SARS-CoV-2 consists of large particles with long spike proteins embedded in a fluid membrane. Accordingly, recruitment of complement is poor, and stimulation of B cells remains suboptimal. A typical VLP with a diameter of ~30 nm with epitopes spaced by 5–10 nm is efficiently recognized by natural IgM, activates the classical complement pathway, enhances B cell activation [31,32] and facilitates deposition of VLPs on follicular dendritic cells, supporting the generation of germinal centers [33] and promoting the formation of long-lived plasma cells resulting in durable antibody responses. As the antibodies induced by the vaccine candidate were of high avidity for the RBD as well as the whole spike protein of SARS-CoV-2, it will be interesting to reveal how this high specificity will affect recognition of RBDs of the newly emerging mutants, such as the British (B.1.1.7), the South African (B.1.351) or Brazilian (P.1) variant. Indeed, we have previously shown that sera from convalescent patients essentially fail to recognize the P.1 variant [34], while the serum from individuals immunized with Pfizer’s BNT162b2 showed 10-fold lower recognition [35]. Thus, cross-recognition of RBDs from mutant viruses may be a critical issue for the future.

Mouse infection with different viruses can trigger an antibody response that is often dominated by the IgG2a subclass. IgG2a shows several better functional properties in comparison to other subclasses [36,37]. These properties include: binding to Fc receptors [38], activating complement pathways [39] and mediating the process of antibody-dependent cell-mediated cytotoxicity [40]. A prominent example are antibodies against the ectodomain of influenza M2 protein, which potently protect mice from influenza virus if produced as IgG2a antibodies but fail to do so if produced as IgG1 [41]. 

We have previously shown that RNA packaged in VLPs is responsible for efficient isotype switch to IgG2a and this response was dependent on TLR7/8 signaling in B cells [42]. Furthermore, bacterial RNA was found to be the most efficient type of RNA for generation of IgG2a, rendering bacterial expression systems optimal for this type of vaccine [43]. Hence, the *E. coli*-based process used here for vaccine production may not only be highly scalable but also results in the most protective antibody responses.

IgA antibodies may be a potent weapon against SARS-CoV-2, as they are expected to neutralize the virus locally on mucosal surfaces without enhancing inflammation via Fc receptors. It is, therefore, interesting to note that we observed induction of strong serum IgA responses by our vaccine candidate. We have previously shown in mice that s.c. immunization with RNA-loaded VLPs results in a systemic IgA response, again dependent on TLR7/8 expression in B cells [44]. This route of immunization did, however, not result in mucosal IgA responses. It remains to be seen whether s.c. immunization with RNA-loaded VLPs results in mucosal IgA responses in humans. 

We have not directly measured induction of Th cells responses by our vaccine candidate, mostly because the expected effector mechanism is induction of neutralizing antibodies. In addition, the size of the RBM domain of roughly 70 amino acids may be too small to reliably induce a Th cell response in inbred mice. However, we have previously shown that VLP-specific Th cell responses mediate isotype switch for B cells specific for antigens displayed on the VLPs. Furthermore, the bacterial RNA packaged in VLPs, such as AP205, drive CD8 and Th1 responses [45,46,47]. Thus, RNA-loaded VLPs and antigens displayed by them induce Th cell dependent isotype-switching and Th1 responses.

Taken together, we show in this study that the bacteriophage-derived AP205 VLPs consisting of dimerized capsid proteins may efficiently be used as a vaccine template for insertion of RBM domain. The resultant vaccine shows high immunogenicity and induced antibodies were neutralizing and able to block virus replication in two independent neutralization assays. 

## Figures and Tables

**Figure 1 vaccines-09-00403-f001:**
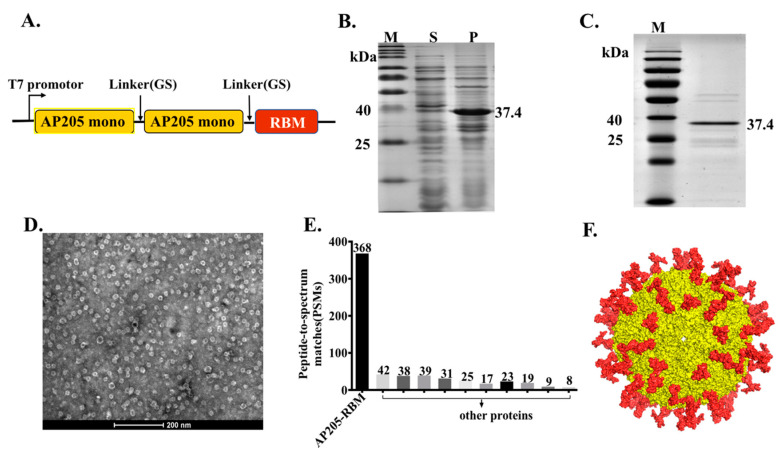
The bacteriophage AP205-VLPs can be used efficiently for generating a fusion vaccine against SARS-CoV-2. (**A**) Schematic representation of AP205-RBM coat protein fusion construct. (**B**) 12% SDS-PAGE for AP205-RBM expression in *E*. coli showing soluble (S) and insoluble (P) fractions. (**C**) 12% SDS-PAGE showing the purified AP205-RBM vaccine of ~37.4 kDa. (**D**) Electron Microscopy of negatively stained AP205-RBM VLPs showing a size of ~30 nm, scale bar 200 nm. (**E**) MALDI-MS of AP205-RBM vaccine. (**F**) 3D model of AP205-RBM vaccine illustrating AP205 CP in yellow and the genetically fused RBM in red.

**Figure 2 vaccines-09-00403-f002:**
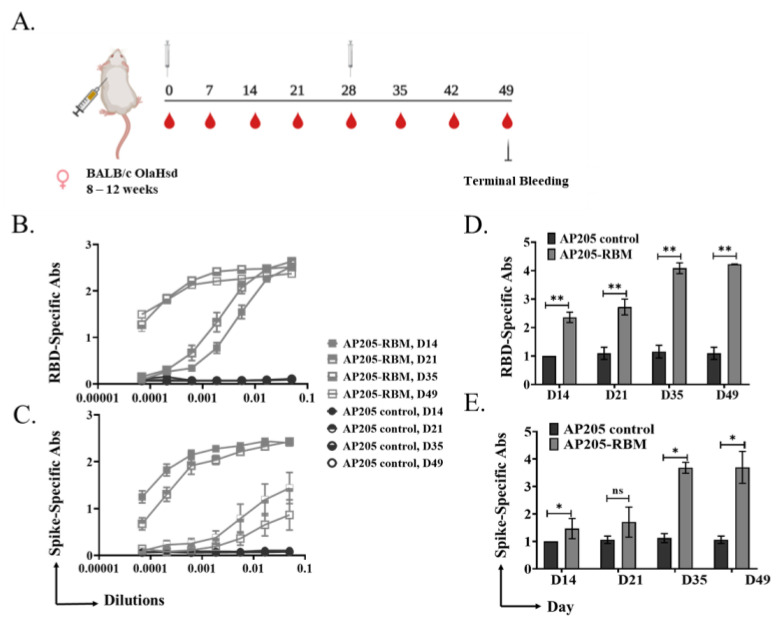
Vaccination with AP205-RBM vaccine induces high titer of RBD and Spike-specific IgG Abs. (**A**) Vaccination regimen and bleeding schedule. (**B**,**D**) RBD-specific IgG titer for the groups vaccinated with AP205 control and AP205-RBM vaccine on days 14, 21, 35, and 49, measured by ELISA (OD50 in D: given as reciprocal dilution values), three-fold serial serum dilution was used starting from 1:20. (**C**,**E**) Spike-specific IgG titer for the groups vaccinated with AP205 control and AP205-RBM vaccine on days 14, 21, 35, and 49, measured by ELISA (OD50 in E: given as reciprocal dilution values), three-fold serial serum dilution was used starting from 1:20. One representative of 2 similar experiments is shown. The value of *p* < 0.05 was considered statistically significant (* *p* < 0.01, ** *p* < 0.001).

**Figure 3 vaccines-09-00403-f003:**
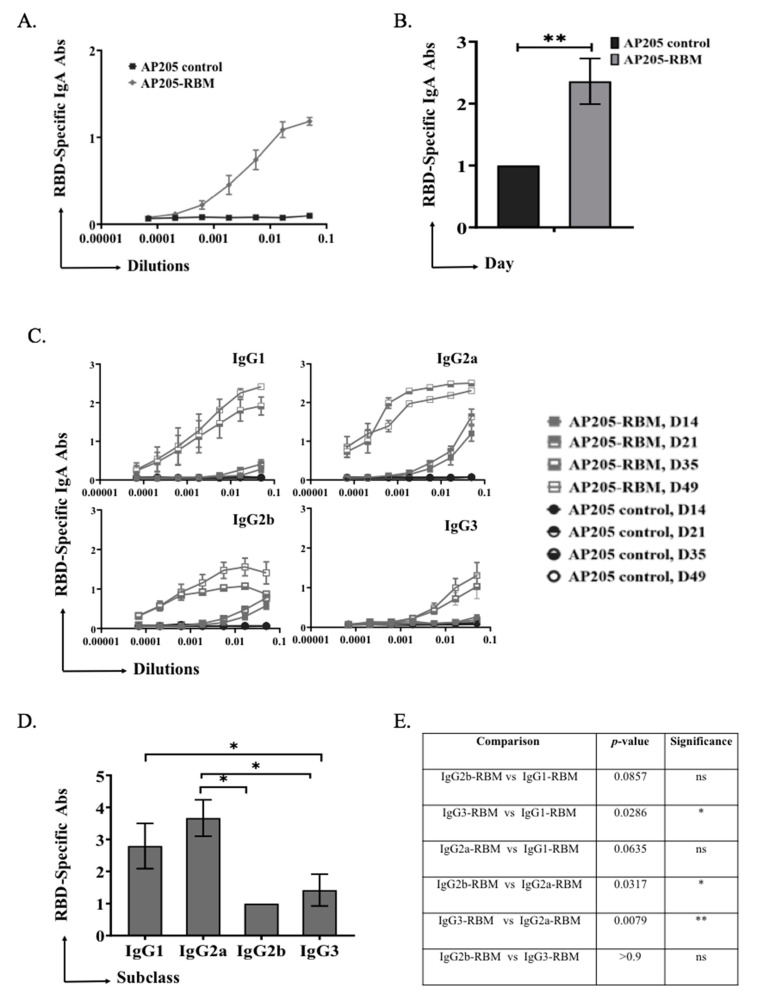
AP205-RBM vaccine promotes IgA production and IgG responses dominated by IgG2a. (**A**,**B**) RBD-specific IgA titers for the groups vaccinated with AP205 control and AP205-RBM vaccine on day 49, measured by ELISA (OD50 in B: given as reciprocal dilution values), three-fold serial serum dilution were used starting from 1:20. (**C**) RBD-specific IgG1, IgG2a, IgG2b, and IgG3 titer for the groups vaccinated with AP205 control and AP205-RBM vaccine on days 14, 21, 35, and 49 measured with OD450, three-fold serial serum dilution were used starting from 1:20. (**D**) RBD-specific IgG1, IgG2a, IgG2b, and IgG3 titers for the group vaccinated with AP205-RBM vaccine on day 49 measured by ELISA (OD50: given as reciprocal dilution values), three-fold serial serum dilution were used starting from 1:20. (**E**) A table summarizing the *p*. value when comparing the different IgG subclasses. One representative of 2 similar experiments is shown. The value of *p* < 0.05 was considered statistically significant (* *p* < 0.01, ** *p* < 0.001).

**Figure 4 vaccines-09-00403-f004:**
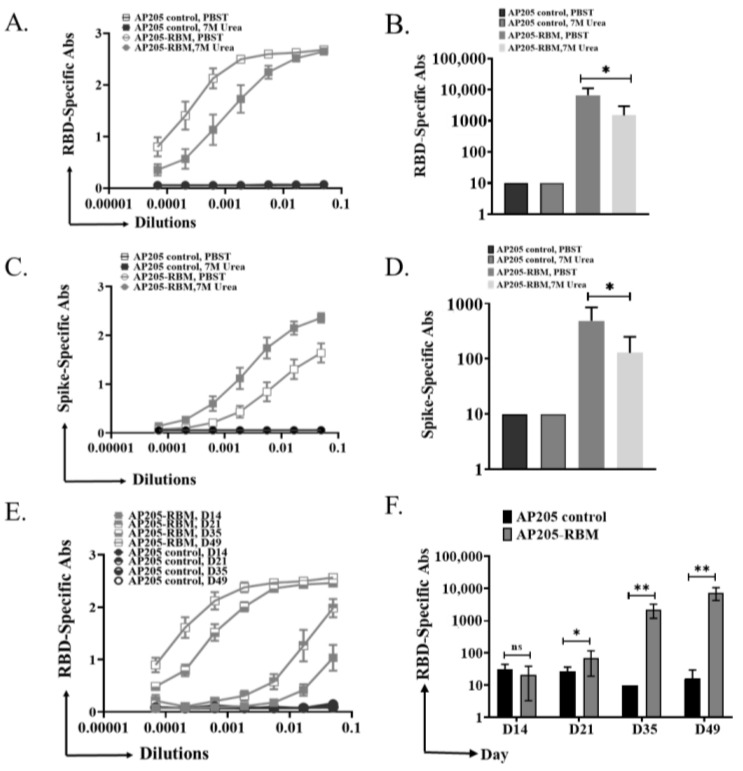
The induced IgG antibodies against RBD and Spike protein by AP205-RBM vaccine are of high avidity. (**A**,**B**) RBD-specific IgG titer for the groups vaccinated with AP205 control or AP205-RBM vaccine using sera from day 49, measured by ELISA (OD50 in B: given as reciprocal dilution values), Plates were incubated in duplicate, after serum incubation one plate was treated with PBS + 0.05% Tween 20 and the other plate with 7 M Urea in PBS + 0.05% Tween 20. Three-fold serial serum dilution was used starting from 1:20. (**C**,**D**) Spike-specific IgG titer for the groups vaccinated with AP205 control or AP205-RBM vaccine using sera from day 49, measured by ELISA (OD50 in D: given as reciprocal dilution values). (**E**,**F**) RBD-specific IgG titer for the groups vaccinated with AP205 control or AP205-RBM vaccine using sera from day 49, measured by ELISA (OD50 in F: given as reciprocal dilution values), Three-fold serial serum dilution was used starting from 1:20. One representative of 2 similar experiments is shown. The value of *p* < 0.05 was considered statistically significant (* *p* < 0.01, ** *p* < 0.001).

**Figure 5 vaccines-09-00403-f005:**
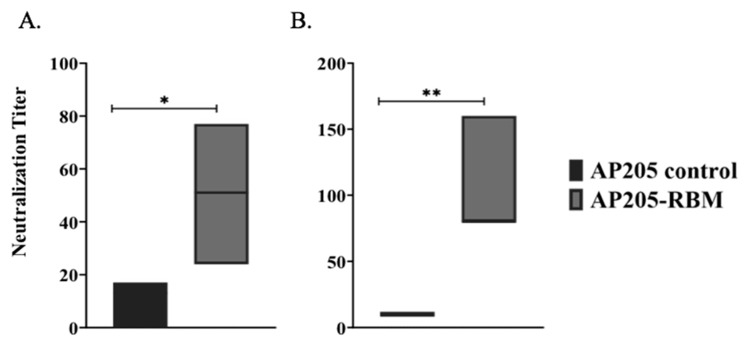
The AP205-RBM vaccine candidate induces antibodies neutralizing SARS-CoV2. (**A**) Neutralization titer of the induced antibodies using a SARS-CoV-2 pseudotyped vesicular stomatitis virus (VSV) assay. (**B**) Neutralization titer of the induced antibodies using CPE method and 100 TCID50 of SARS-CoV-2/ABS/NL20 virus. One representative of 2 similar experiments is shown. The value of *p* < 0.05 was considered statistically significant (* *p* < 0.01, ** *p* < 0.001).

## Data Availability

The data presented in this study are contained within the article.

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
