# Peer review of "AP205 VLPs Based on Dimerized Capsid Proteins Accommodate RBM Domain of SARS-CoV-2 and Serve as an Attractive Vaccine Candidate"

_vaccines, 2021, doi:10.3390/vaccines9040403_

Round 1

Reviewer 1 Report

Authors developed AP205 VLPs-based vaccine against SARS-CoV-2. Under sever pandemic by Covid-19, it is urgently needed for pharmaceutical drugs or vaccines against SARS-CoV-2. Goal of vaccination is to induce acquired immunity including humoral and cell-mediated immunity. Especially, the humoral immunity is important to prevent host cells from infection. The manuscript is well written, and experiments are also organized. But there are some points remained to be assessed.

CD4+ helper T-cells would play a critical role for the immunoglobulin class switching. Does the AP205-RBM vaccine stimulate spike-specific T-cell responses, especially, Th cells in the vaccinated mice? Please show some data or discuss that.

Recently, mutated strains of SARS-CoV-2 has found and cause problems about less efficiency of vaccination developed based on original strain. The AP205-RBM contains point mutated amino acid in the “England mutated strain”. Induced antibodies by the vaccination would be effect against them? Please discuss.

Minor points

There is a detailed method of experiment described in result section. Please describe more briefly.

Figure 4 A and C, symbols and their legends in graphs appear to be incorrect. Please make a correction.

Also, figure 4E, there is no legends. which symbol is which sample?

Author Response

  1. CD4+ helper T-cells would play a critical role for the immunoglobulin class switching. Does the AP205-RBM vaccine stimulate spike-specific T-cell responses, especially, Th cells in the vaccinated mice? Please show some data or discuss that.

 R: We now discuss the role of Th cells and that VLPs loaded with RNA induce Th1 and CTL responses.

  1. Recently, mutated strains of SARS-CoV-2 has found and cause problems about less efficiency of vaccination developed based on original strain. The AP205-RBM contains point mutated amino acid in the “England mutated strain”. Induced antibodies by the vaccination would be effect against them? Please discuss.

R: We add this now to the discussion. Thank you for raising this point.

  1. Figure 4 A and C, symbols and their legends in graphs appear to be incorrect. Please make a correction. Also, figure 4E, there is no legends. which symbol is which sample?

R: Thank you. We made a correction on Figure 4 and added symbol for samples in figure 4E.

Reviewer 2 Report

The study describes the production of virus-like particles (VLPs) constructed to display the SARS-CoV-2 receptor-binding motif and subsequent laboratory studies of immunogenicity.  The study is written and presented to a high standard.  My comments by manuscript section are as follows.

Title.  Appropriate to the work presented.

Abstract.  Appropriate to the work presented. 

Introduction.  Appropriate to the work presented.  ? missing reference lines 40-41.

Materials and methods.  Comprehensive and sufficiently detailed so that the work could reasonably be repeated by others.  Please define OD50.

Results.  Comprehensive, sufficiently detailed, appropriately structured, and relevant to the study undertaken.

Discussion.  Relevant and appropriately referenced.  Some additional and detailed comments on the potential impacts of spike protein mutations and virus variants would be useful.

Author Response

  1. Appropriate to the work presented.  ? missing reference lines 40-41.

R: This has been fixed.

  1. Materials and methods.  Comprehensive and sufficiently detailed so that the work could reasonably be repeated by others.  Please define OD50.

R: Thank you. We define OD50 as ’ Antibody titers were determined as the concentrations of antibody at 50% maximum optical density (OD50). ’

  1. Relevant and appropriately referenced.  Some additional and detailed comments on the potential impacts of spike protein mutations and virus variants would be useful.

R: This has been added. Thank you for raising this relevant point.